# 'Where have all the doctors gone?' A protocol for an ethnographic study of the retention problem in emergency medicine in the UK

Daniel Darbyshire [1,2] Liz Brewster [2] Rachel Isba [2,3] Richard Body [4,5] Dawn Goodwin [2]

[1]Emergency Department, Salford Royal NHS Foundation Trust, Salford, UK
[2]Lancaster Medical School, Lancaster University, Lancaster, UK
[3]Children's Accident and Emergency Department, North Manchester General Hospital, Manchester, UK
[4]Division of Cardiovascular Sciences, The University of Manchester, Manchester, UK
[5]Emergency Department, Manchester University NHS Foundation Trust, Manchester, UK

**Correspondence to**
Dr Daniel Darbyshire;
d.darbyshire@lancaster.ac.uk

## ABSTRACT

**Introduction** 'Emergency medicine (EM) in the UK has a medical staffing crisis.' Inadequate staffing, in EM and across healthcare, is a problem that affects the quality of patient care globally. Retention of doctors in EM is a particularly acute problem in the UK's National Health Service. Sustainable careers in healthcare are gaining increasing attention at a national and international policy level, but research to understand the factors that facilitate retention is lacking.

This study aims to develop understanding of what drives retention of doctors in EM by focusing on those who remain in these careers, where previous research has targeted those who have left. By addressing the problem of retention in a different way, using innovative methods in this context, we aim to develop a deeper and more nuanced understanding of sustainable careers in EM.

**Methods and analysis** This is an ethnographic study combining participant observation in two emergency departments, interviews with doctors from these departments, from organisations with influence or interest at a policy level and with doctors who have left EM. The analyses will integrate detailed workplace observation alongside key academic and policy documents using reflexive thematic analysis.

**Ethics and dissemination** Approvals have been obtained from Lancaster University via the Faculty of Health and Medicine Research Ethics Committee (FHMREC18058) and the Health Research Authority (IRAS number 256306). The findings will inform understanding of sustainable careers in EM that may be transferable to other settings, professions, and locations that share key characteristics with EM such as paediatrics, emergency nursing and general practice. Findings will be disseminated through a series of academic publications and presentations, through local and specialty research engagement, and through targeted policy statements.

## Strengths and limitations of this study

► Focuses on a national research priority and a global problem.
► Takes a novel approach to understanding the retention problem.
► Engaged with policymakers and the public throughout the research.
► Some aspects of retention will vary by context.
► Policies and practice will evolve during the research, making it harder to unpick some of the findings.

emergency department (ED) visit concluded within 4 hours, and a rise in numbers of patients spending greater than 12 hours in the ED.[1] Much of this declining performance is attributed to factors outside of the ED, with exit block due to insufficient hospital beds and community services being the most frequently cited contributing factors.[2 3] Arguably, the most important factor for high-quality care in the ED that departments have some control over, is staffing.

Unfortunately, 'EM in the UK has a medical staffing crisis.'[4] Starting with the formation of the Emergency Medicine Taskforce, set up in 2011,[4] the specialty has sought to change practice, policy and culture through innovations such as the 'less than full time training' pilot,[5] increasing recognition of the particular stresses a career in EM brings and a focus on sustainable careers.[6 7] This has led to EM growing rapidly in terms of both consultant and trainee numbers but demand for care has outstripped staffing growth and looks to continue to do so.[8]

The EM workforce problem is occurring in the context of staffing shortages in much of the UK's National Health Service (NHS)[9] and on a background of demand outstripping growth in the global healthcare workforce.[10] The picture is complex. The NHS employs

## INTRODUCTION

The specialty of emergency medicine (EM) provides care across the spectrum of illness and injury to people of all ages. It operates all day, every day and in the UK has seen demand increase year on year, with declining performance in terms of patients having their

around 1 million people, with the workforce increasing by around 1.8% from July 2017 to 2018. However, organisations report collective vacancies across the NHS in the region of 100 000 and falling numbers of key staff groups.[9] The *NHS Long Term Plan*,[11] published in January 2019 (updated August 2019), followed by the *Interim People Plan*,[12] outlined the UK government's proposed solutions for staffing the NHS. The plans—which focused on staff development, empowerment and improving leadership culture—were broadly welcomed.[13] There were, however, concerns, for example about the time lag between increasing training numbers and generating qualified practitioners in employment, with potential over-reliance on international recruitment in the interim.[13 14]

### From recruitment to retention

EM historically had a recruitment problem, but after much work by the then College of Emergency Medicine (now the Royal College of Emergency Medicine) and Health Education England, among others, in improving the image of the specialty and the experience of trainees, fill rates for EM training posts reached 96% in 2011.[15] Recruitment has since fallen from the 2011 high, yet it remains strong and competitive, when compared with other specialties, at 86% in 2019 (compared with 77% for internal medicine and 69% for paediatrics).[16] However, attrition from training programmes is unsustainably high, with The King's Fund (an independent English think tank) reporting that: 'problems with retention mean it (emergency medicine) has the greatest attrition rate of any medical specialty, with almost 50% of registrar doctors in their third or fourth year of training resigning.'[17]

Following on from a 2014 survey of trainees that looked at the attractiveness of a career in EM,[18] the Emergency Medicine Trainees' Association now conduct its own annual surveys.[19–21] These surveys elucidate the challenges faced by trainees, specifically burnout, poor working conditions and workload pressures. This is supported by the General Medical Council's national training survey which showed EM as the specialty with the highest work intensity[22] and by the high attrition from the EM specialty training programme.[23] The data on trainees are reflected for consultants and staff and associate specialist grades, with all groups being undersubscribed and experiencing exodus.[23]

Recruiting and retaining EM clinicians is fundamentally connected to the provision of high-quality clinical care. More experienced and senior doctors deliver better care, performing fewer unnecessary tests and getting fewer complaints, and experience is only gained by retaining doctors within the specialty.[24–26] Retention of doctors affects wider health and care services in two main ways. First, the presence of doctors with more experience results in efficiency savings both in the ED and downstream in a patient's hospital journey, with recent studies finding a greater than 20% reduction in admission rates, reduced waiting times and cost savings of £3 million to one hospital.[24 25] The introduction of the Major Trauma

Networks in 2010 with consultant-led care of severely injured patients in London saved an estimated 58 lives in the first year.[26] Second, staff turnover is costly and disruptive. When ED staff leave it is highly unusual for replacements to be found quickly, if at all, often leading to locum (temporary) doctors filling in who are 'generally less efficient, and less safe, than permanent members of staff'.[27] Costs in the UK are not clear but an approximate figure for replacing one fully trained emergency physician in the USA is $164 000.[28]

We would argue that the evidence is therefore clear that staff retention is important for quality emergency care. What is not clear is what drives retention and what allows people to have sustainable careers. This knowledge gap was also identified in the 2017 James Lind Alliance (see figure 1) priority setting partnership for EM research. Number 4 in their list of the top 10 research priorities in EM asked: 'with regard to how ED staff development is managed, what initiatives can improve staff engagement, resilience, retention, satisfaction, individuality and responsibility?'[29] This priority was supported by calls from various other stakeholders to address the issue of retention including The King's Fund,[17 30] Health Education England[15] and the Royal College of Emergency Medicine.[27 31–33]

The proposed study aims to address retention of doctors in EM—an innovative approach, contrasting with previous work that has focused predominantly on why people leave. The ethnographic methodology is also innovative in this setting as the retention problem has not previously been explored using this approach. By developing fresh insights into the complex and interacting factors that positively influence retention, we hope to inform future efforts aimed at reducing exodus from the specialty, at both a research and policy level.

### Research aim and objectives

The aim of this study is to gain a deep understanding of retention in EM, in order to elucidate how retention is made possible. This aim requires a study design that can tackle complexity and develop new understandings. The objectives are to:

► Understand in detail the day–to–day lived experience of EM doctors, to identify and explore factors influencing retention.
► Situate these descriptions within the current educational and health policy contexts.
► Advance the debate and make policy and practice recommendations based on a detailed understanding of retention of doctors in EM.

### METHODS

### Overview

We will conduct an ethnographic study of EM doctors in two type 1 EDs in the North West of England. Ethnography is derived from anthropology and social sciences and is historically associated with gaining cultural understandings about groups, with the origins of the practice going

## The James Lind Alliance

The James Lind Alliance (JLA) is a non-profit making initiative established in 2004. It brings patients, carers and clinicians together in Priority Setting Partnerships (PSPs) to identify and prioritise the Top 10 unanswered questions or evidence uncertainties that they agree are the most important.

The aim of this is to make sure that health research funders are aware of the issues that matter most to the people who need to use the research in their everyday lives. It is part of the NIHR (National Institute for Health Research), the research funding arm of the United Kingdom's NHS (National Health Service).

**Figure 1** What is the James Lind alliance? information taken from the JLA website; http://www.jla.nihr.ac.uk/about-the-james-lind-alliance/ (accessed 15th January 2020).

back to colonial-era studies of 'exotic' civilisations.[34] Its use within health research is broad and well established, from a 1961 study of medical student culture[35] through to a recent study trying to understand why one particular maternity unit performs highly.[36]

Ethnography is 'a systematic approach to learning about social and cultural life of communities, institutions and other settings'.[37] Key features of ethnography relevant to this study are that it is investigative, the primary tool of data collection is the researcher, and that it 'emphasises and builds on the perspective of the people in the research setting'.[37]

Ethnography creates understanding through description and analysis. This usually involves going to a place and collecting data—which is often a detailed description of what happens in that place along with the accompanying talk—over a period of time. What information is recorded, how long the researcher spends there and what other forms of data are collected, depends on the context of the research and question being asked. For this research, ethnography refers to the totality of the study including the time spent observing in the department (referred to as participant observation hereafter), interviews and policy documents.

This ethnographic study will involve the lead researcher spending time in two different EDs, trying to understand what allows doctors to have sustainable careers. This will involve observing the people, space and happenings in the departments, and speaking to those in these spaces, keeping the focus on retention. Without aiming to predict retention-related behaviours, this may mean observing humour, conversations between members of staff or how they work within the challenging working environments. This will be supported by interviews with

doctors (conducted away from the department) and with others who can inform the research aim and objectives, and by critical review of policy and academic literature relating to retention. The literature search strategy, in the form of a scoping review, is described elsewhere.[38] Each step in the process is expanded on below and summarised in figure 2. The study is inherently iterative, observations and interviews will inform subsequent observations and interviews, and the study plan may adapt to analysis, and opportunities and challenges in the field.

This study subscribes to the 'ethnographic paradigm',[39] by this we mean that we do not see value in predetermining a particular way of seeing the world, and agree with Atkinson when he writes that 'ideas can and should come from a variety of sources'.[40] This study draws on the sociological practice of ethnography and as such the findings will be analysed, interpreted and challenged through a number of sociological theories. These will include, but are not limited to, theories of work[41 42] such as emotional labour[43] and social capital,[44] theories of community such as Tönnies discussion of Gemeinschaft and Gesellschaft (community and society),[45] and Bourdieu's conception of symbolic power.[46] This list is not intended to be prescriptive—it aims to show theories that may prove informative.

As the study is built on the interpretation of DD (the researcher collecting the data and performing the analysis), a discussion of their positionality will be included when the study reports.[47] This is an integral part of reflexive thematic analysis described below, 'addressing threats to credibility in ethnography requires different techniques from those used in experimental studies.'[48] Our study aims to produce a clear audit trail, of which publishing this protocol is a part.[38 49] Other important parts of the audit trail are notes made during observations

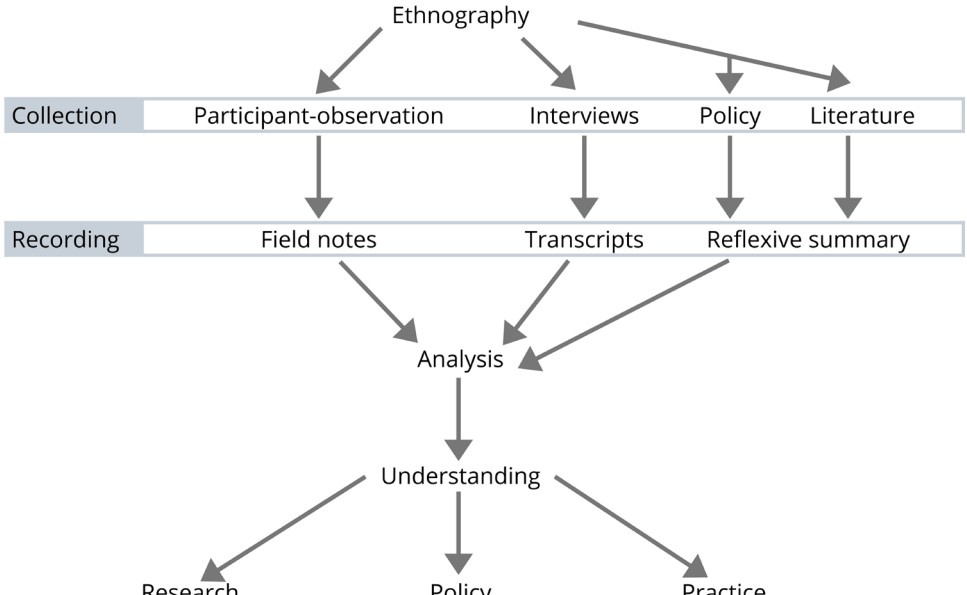

**Figure 2** Summary of the ethnography. The arrows suggest a linearity to the study which is not reflected in the reality of ethnographic practice, instead the image is aiming to convey the different workstreams that come together in the analysis. The figure also fails to include the messy connections between the different parts of the study and the overlap between the intended outputs.

about why the researcher has chosen to observe at particular times and locations, documents generated around meetings of the research team, and documents from peer-review, ethics and the funder which have shaped the study.[49] We are aiming for transferability not generalisability, reflecting the importance of context,[50] and will use reflexive thematic analysis, and sense checking from the public involvement group. We aim for confirmability by aligning interpretations and data in the final report.[51]

### Patient and public involvement

The public has been involved in the development of this study at several levels and at different key points in the development of the protocol. The aforementioned James Lind Alliance priority setting exercise demonstrated the public's belief that the topic areas are important.[29] To confirm this, with the support of the National Institute for Health Research (NIHR) Research Design Service North West and the Research and Development Department at Blackpool Victoria Hospital, we ran a public involvement event. During this event, members of the public agreed with the James Lind Alliance that exploring factors related to the retention of doctors in EM merited investigation. Participants thought the proposed methods were acceptable to patients, but questioned whether ethnography would be seen as acceptable to a clinical audience. A member of the initial patient involvement group has volunteered to be further involved in the study by becoming a member of the steering committee and by reviewing the conclusions we gather during the scoping review part of the study.[38]

As the main focus of the study is EM doctors, we also sought their views as part of our engagement strategy. We presented the research plan, along with some of the preliminary findings from the scoping review,[38] at the Emergency Medicine Trainees' Association National Conference in Cardiff, in November 2018. Feedback was obtained during the question and answer portion and by providing a link to an online form. This process was repeated at the Oxford Society for Emergency Medicine Conference and the Association for the Study of Medical Education Researching Medical Education Meeting in London, both in December 2018. Thirteen attendees provided written feedback, including one offer to help, which led to recruitment of a second reviewer for the scoping review, with many more providing verbal comments and questions. Including interviews with doctors who have left was suggested during this professional engagement, and again during peer review from the NIHR when funding was being obtained, and subsequently included in the research plan.

### Participant observation

Participant observation refers to the time spent in the research setting watching, listening and documenting. It is strongly linked with ethnography and is sometimes synonymous with field work. These terms are contentious[52] and for clarity we refer to participant observation as the component of ethnography that occurs in the research setting. All data collection will be conducted by DD who will undertake observation at two sites, each for a period of 12 weeks. At each site they will perform observations in all parts of the department and at all times of the day. A day of participant observation will be divided in two. The first half of the day will be spent conducting observations and writing field notes, incorporating talking with staff and documenting what they say, and the researcher's thoughts about the encounter.

We will investigate the working environment, interaction between staff and behaviours related to retention. Clinical encounters are not the focus of this study, they have been studied from multiple perspectives and are well understood by the primary audience for this research–emergency physicians and EM leaders. The field notes will be handwritten in a notebook. The other half of the day will involve typing up, reflecting on and expanding field notes. We aim to perform 4 days of observation each week by scheduling 48 days over the 12-week period with start times reflecting the shift patterns locally, but to include days, evening, nights and weekends. A total of 24 weeks' participant observation was chosen as it will allow DD to participate in observations at all times, with flexibility to repeat observations at key times or with specific key informants. Traditionally ethnographic observation has been conducted over a much greater period of time than the one that proposed, to allow for a period of familiarisation. This process will be accelerated in this study due to DD's pre-existing knowledge of EM and EDs (as a senior trainee in EM). This more focused form of ethnography has a developing literature for its use in complex healthcare settings[53] and based on this work, and the intensity of the observation within the 12-week period, it is estimated that this will be sufficient to meet the aims of the study.

## Interviews

In addition to the participant-observation conversations described above, we will conduct interviews, reserving the term for the physically separate, planned, audio-recorded and transcribed encounters. DD will conduct all the interviews, aiming to complete around 40. Ten interviews with doctors across all levels of seniority from each of the two research sites, 10 interviews with individuals within key organisations relevant to retention in EM (such as the Royal College of Emergency Medicine, the Emergency Medicine Trainees' Association, Health Education England and NHS Improvement), and 10 with doctors who have left EM. Potential participants for interviews will be identified during the periods of participant observation and the literature review and supplemented by recommendations obtained from interviewees. Interviewing 10 doctors from each research site would represent between a quarter and one-third of doctors at each level present on a rota at any time. The sample size has been agreed on by discussion within the research team, who have extensive experience with this methodology, and by consulting key texts, and represents an estimation for when theoretical saturation (see the Glossary section) might be achieved.[54–56]

Interviews will take place in a neutral space which allows for private conversation. The interview guide is available in online supplemental appendix 1. However, it is expected to evolve as the study progresses, based on pilot interviews and ongoing analysis. Interviews will be digitally recorded, transcribed and imported into NVivo V.12,[57] a qualitative data analysis computer package that facilitates the organisation and analysis of research data. After each interview, DD will record reflections in the field notes.

## Policy and literature

Policy documents (grey literature) will be identified through the aforementioned scoping review[38] and from the other aspects of the ethnography. Key documents will be chosen based on relevance to research context and incorporated into the analysis. A similar process will be used for academic literature.

## Inclusion criteria

Participant observation sites were selected based on the following factors:
▶ Presence of foundation, core and higher trainees.
▶ Presence of staff and associate specialists.
▶ Type 1 ED (major ED, providing a consultant-led 24-hour service with full resuscitation facilities).[58]

### Interview and observation candidates

Within the two EDs, the focus for participant observation and interview candidates is:
▶ Doctor.
▶ Working in the study site.
▶ Staff member for at least 4 weeks.
▶ Aim to include a maximally inclusive representation of the department, including age, gender, seniority, international medical graduates and UK-trained doctors.

The inclusion criteria are intended to be broad and inclusive. The 4-week criteria were chosen to avoid interviewing very new staff members at the start of their work. The longitudinal nature of the ethnography means that these candidates may be eligible once they have been in the department for a month or more.

## Consent
### Participant observation

Informed consent needs to match the research and the environment in which it is being undertaken. We are not alone in making this point, several previous studies within the NHS[36 59] and the ED[60] have taken a proportionate route, clearly demonstrating the acceptability and utility of such an approach.

To ensure participants are informed of the study protocol, aims and data collection methods, a continuous information-sharing approach will be taken, mirroring the studies referenced above. The research has already been explained to the senior leadership at each study site, along with the research and development leads. To complement this, we will present the research plan to the medical and nursing staff at scheduled meetings such as junior doctors teaching, band 5, 6 and 7 nursing meetings, consultant meetings and departmental research and governance meetings. We will advertise the start of the study with posters giving people the opportunity to ask questions and state objections, before it starts. Emails will be circulated to all ED staff informing them of the study. In addition, we will offer a brief explanation to staff members who the researcher interacts with in the department, as both part of the consent process and as a way to start conversations as part of the research method.

When introducing himself in the research setting, DD will reiterate the voluntary nature of the study and that it is okay for them to ask for observation to stop, either temporarily or until further notice. Based on previous studies in the ED this is unlikely to occur frequently, and as such we intend, if this happens, to relocate to a different part of the department to continue observation. If the objection is not time limited then the researcher would not attend when that person was on shift, the 24-hour nature of EM makes this feasible. Should an individual ask for more information about the study, we will provide them with a short summary of the study, a link to this protocol and contact details for the research team.

Patients are not participants in this study, and no patient identifiable data will be recorded, but they, and other staff members, do occupy the research space. Their confidentiality and acceptance of the research is vital. We believe it is unfeasible, overly burdensome and unnecessary to obtain consent from patients transiting through the ED. Either DD will introduce themselves or will be introduced by the member of staff they are accompanying. If a patient does not want to have the care they receive observed, DD will withdraw from that episode of care. DD will rely on his experience and expertise of working within this environment to proportionally address this issue in real time.

Informed consent for ethnographic study is different to other types of research, particularly so in unpredictable settings such as the ED. The use of the standard written consent procedure is overly burdensome to the researcher and participant, not feasible, and would bar patients and staff members who occupy these spaces from benefiting from this type of research.[61]

### Interviews

Formal written consent, supported by a participant information sheet, will be obtained for the formal, recorded interviews. Consent will be obtained by DD prior to conducting the interview. Having expressed initial interest in participating, potential interviewees will be provided with a copy of the consent form and participant information sheet. On the day of the interview, the interviewee will be given another opportunity to review the participant information sheet and consent form, with DD on hand to answer any questions. Some interviewees, especially those with high-profile roles in national organisations, may feel that anonymisation is either impossible or not in their interests; they will be given the option of participating in the study in a non-anonymised fashion. They will retain the right to have their interview anonymised, and all interviewees have the right to withdraw, up to the point of publication.

### ANALYSIS

During the data collection phase of the study, initial analysis will take the form of reflective analytical notes recorded in the field notes, these observations and interpretations guide the ongoing analysis. Once data collection is complete, a more thorough, systematic analysis of field notes, interview transcripts and literature and policy documents will be completed using reflexive thematic analysis.[62–64] This process leads to generation of themes—these are not groups of data that are about the same thing, but 'stories about particular patterns of shared meaning across the dataset'.[62] Reflexive thematic analysis emphasises the effortful nature of analysis, in that themes are not found within the data, rather they are created though analytical work. This highlights the role of the researcher in generating the analysis, laying this open to scrutiny. The analysis will use the six-stage approach to thematic analysis (see figure 3 for a summary).[62 65] These stages are descriptive and supportive, as opposed to having clear divisions. It is expected that the analysis will go back and forth through the stages multiple times.

### Limitations

The study is geographically limited to the North West of England. Much of the activity of EDs is similar nationally, but local differences will mean the findings will require interpretation and translation to other locales.

The study is based at two sites. There is no reliable objective way of stating if the sites have high or low retention, relative to the national picture. If they are extreme outliers it may be difficult for other sites to interpret the findings.

This study is focused solely on doctors working in EDs, there may be some transferability to other professions or working environments, but this is a limitation.

Data collection by someone without experience of the ED may lead to different conclusions, though this might be at the expense of access problems and detailed understanding of the work in the ED.[52]

### ETHICS AND DISSEMINATION

Approval from Lancaster University via the Faculty of Health and Medicine Research Ethics Committee (FHMREC18058) was received on 15th April 2019. Health Research Authority approval was formalised on 11th November 2019, IRAS number 256 306. Traditional academic dissemination will occur locally through seminars and lectures, and to a broader audience through conference presentations and academic publications. Dissemination to the public will be facilitated by presentations at knowledge translation events and by engaging with local media and research translation services such as The Conversation. Policy briefs summarising the findings will be provided for key audiences such as NHS England, Health Education England and the Royal College of Emergency Medicine.

### GLOSSARY

Audit trail—a thorough collection of documentation regarding all aspects of the research. Qualitative research often evolves through the iterative process of data collection and analysis, requiring the researcher to make frequent decisions which alter the course of the research.

| Stage | Description | Applied to this study |
|---|---|---|
| 1 | **Familiarisation** – involves becoming immersed in the data, getting to know it. It is about noticing things that may (or may not) be important. | This will involve reading field notes, transcripts, and other documents, and making notes about connections, possibilities, and quirks. |
| 2 | **Generating codes** – attaching labels known as codes to portions of data to create meaning at the intersection between the data set and interpretative resource that is the researcher. | Using qualitative data analysis computer software (NVivo 12 for Mac) to attach codes to data. Coding choices will be documented and attached to each code. |
| 3 | **Generating initial themes** – a good theme both identifies the area of data and tells the researcher something about it. These themes will evolve or be discarded. Sometimes called candidate themes. | Stepping back from the fine detail of the data to try and group codes that inform a specific issue. |
| 4 | **Revising/reviewing themes** – involves reviewing themes against the coded data and the entire data set. Mismatches lead to revisions. | Compiling all data for each candidate theme and reviewing against the coded data and data set in turn. |
| 5 | **Defining and naming themes** – creatively attaching names to themes that signal the core of theme. This aims to move analysis to the conceptual level while maintaining links to the data. | Thinking about the audience and research objectively to name themes that can help effectively communicate the findings. |
| 6 | **Writing** – producing the output from the research. The writing tests the themes. How well do they work together and in isolation? How do they stand up to the literature? These challenges lead to returning to earlier analytical stages. | Writing up the doctoral thesis, papers for medical and sociological audiences, and producing summaries for policy makers and the public. |

**Figure 3** The six stages of thematic analysis as applied to this study. An iterative process, it is expected that the analysis will go back and forth between stages a number of times.[63 66]

Documents that form this trail of evidence include field notes, a research diary and correspondence.[49 66]

Ethnography—a method based on description and the key assumption that 'personal engagement with the subject is the key to understanding a particular culture or social setting'.[67] It can involve many different ways of collecting data, but is strongly associated with participant observation and field work.

Participant observation—a key tool in ethnographic investigation and in effect a combination of a wide variety of methods including observation, informal interviews and conversations, combined with analysis of other materials encountered in the research setting. The term describes the two activities that the researcher does, to a greater or lesser extent, to gain access and trust in the field site and to develop understanding of what is happening there.[68]

Positionality—how the researcher is positioned with respect to the social and political context of the study. Affects all aspects from formation of the research question to analysis. Making this explicit allows the reader to interpret the findings in context.[47 69]

Reflexivity—conscious examination and explanation of how the researcher's positionality influences and is influenced by the research as it progresses.[70]

Theoretical saturation—the continuation of sampling and data collection until no new conceptual insights are generated. At this point the researcher has provided repeated evidence for his or her conceptual categories.[71]

Trustworthiness—the ways in which a qualitative researcher ensures quality in their research, specifically in terms of transferability and credibility.[51] Moves discussions about research quality away from the quantitative paradigm of triangulation and generalisability.

**Acknowledgements** The authors would like to acknowledge the ongoing work of Gavin Quick and Anne Clark for their public and patient involvement role in developing the protocol.

**Contributors** DD conceived the project. DD, LB, RB and DG developed the initial proposals for funding. DD, LB, DG, RI and RB developed the protocol. DD produced the initial draft of the manuscript. LB and DG are providing direct PhD supervision for DD. RB and RI are supporting the PhD with specialty and institutional expertise, respectively. All authors contributed to and revised the final manuscript.

**Funding** DD is funded by a National Institute for Health Research (NIHR) Doctoral Fellowship for this research project. Initiation of the study was supported by the BMA Foundation Kathleen Harper Award and the RCEM Young Investigator Award.

**Competing interests** This publication presents independent research funded by the National Institute for Health Research (NIHR). All authors have completed the ICMJE uniform disclosure form at www.icmje.org/coi_disclosure.pdf and declare:

DD had financial support from the NIHR, BMA Foundation and the Royal College of Emergency Medicine for the submitted work.

**Patient consent for publication** Not required.

**Provenance and peer review** Not commissioned; externally peer reviewed.

**ORCID iDs**
Daniel Darbyshire http://orcid.org/0000-0001-5619-0331
Liz Brewster http://orcid.org/0000-0003-3604-2897
Rachel Isba http://orcid.org/0000-0002-2896-4309
Richard Body http://orcid.org/0000-0001-9089-8130
Dawn Goodwin http://orcid.org/0000-0002-9435-9107

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
