## [Reviewer comments · BMJ Open]

ARTICLE DETAILS

TITLE (PROVISIONAL)	'Where have all the doctors gone?' A protocol for an ethnographic study of the retention problem in Emergency Medicine
AUTHORS	Darbyshire, Daniel; Brewster, Liz; Isba, Rachel; Body, Richard; Goodwin, Dawn

VERSION 1 – REVIEW

REVIEWER	Jenny Advocat Monash University, Australia
REVIEW RETURNED	19-Mar-2020

GENERAL COMMENTS	Thank you for the opportunity to review this protocol. It is a sound and very interesting study. I have only minor comments, suggestions and a couple of questions. I'll work sequentially through the paper from the introduction. INTRODUCTION In the third paragraph you talk about a broad issue in staffing, but when I read statistics about GPs, it seemed to muddy the waters for me. I think it would be more clear to stick to what's happening in ED, unless you are making a case that the findings of this study may have implications for staff retention in General Practice. Line 125 should have a reference after "specialty"—although I see the following sentences are referenced, it would be helpful to also reference that broad claim. METHODS I am curious about your use of "participant" observation. This is indeed the traditional term but I think what you are describing is non-participant observation. I see that the researcher doing the data collection, the ethnographer, is also an ED doctor, unless they are also acting as a doctor in this setting during observation, I would have thought the term was non-participant observation. You note further down (lines 205-6) that an ethnographic approach is open-ended, but on line 197 you say that speaking to doctors in ED, the focus will be kept on retention. This struck me as not very ethnographic because at the heart of ethnography is the idea that you might go in to a setting/culture and find connections that you didn't expect, that might not jump out at you at first, so it's important to keep the inquiry broad, even with tight timelines. Perhaps it's just something to consider. I think it also becomes really clear that people often say one thing and do another, and this is found in ethnographies everywhere!
--

	In line 240 you use the term “sense-checking” and I wondered if that is a particular method or just a phrase? If the former can you please reference and expand? Or perhaps reword for clarity? INTERVIEWS You mention on line 296 “pilot interviews” but I didn’t see them mentioned above, did I miss it? Will these pilots be an early subset of participants outlined above, or others? TYPO: line 314 “interviews candidates”- “s” at end of interview should be deleted I was interested that the inclusion criteria is “staff members for at least 4 weeks” given the introduction describes the benefits of experienced staff. I wonder why the criteria wasn’t for staff who had been working for much longer so they could reflect on actual experiences, rather than expected futures? Line 354, It’s not that clear what is meant by this sentence, starting with “Capacity...” As in staff members might not have full faculties (as in be unconscious?)? Seems so unlikely it feels odd to mention. TYPO? Line 355 “is” at the end of the line should read “in” Regarding consent, it does seem very unlikely but I wonder if there are plans for if a staff member is opposed to ever being observed and how that will impact general observation in that facility? This isn’t an ethics application so it might not need to be included here, but it might be worth noting as it’s high stakes for this study. Again, it’s not an ethics application but in my experience it is not possible to allow people to withdrawal from the research all the way up to publication (line 384) since their data has already long before been collated and used to form the basis of analysis. Aside from these minor notes I think this protocol will add valuable knowledge about staff retention in ED and I look forward to reading the results after all the hard work described here. Best of luck.
--	--

REVIEWER	Dr Katie Gillies University of Aberdeen, United Kingdom
REVIEW RETURNED	29-Mar-2020

GENERAL COMMENTS	Where have all the doctors gone?’ A protocol for an ethnographic study of the retention problem in Emergency Medicine Bmjopen-20202-038229 Darbyshire et al Thank you for the opportunity to review this manuscript titled ‘Where have all the doctors gone?’ A protocol for an ethnographic study of the retention problem in Emergency Medicine. The manuscript reports the protocol for the exploration of staff retention in emergency medicine. Overall the protocol is well written and describes an interesting study with the potential for key learning points. However, I believe there are some aspects that require attention before publication. Specific queries are listed below. Abstract 1. Throughout the abstract it would help the reader if when mentioning retention that the term staff I also included. Also, here
--

	and throughout the protocol to would be helpful to highlight that the staff being referred to are doctors and not other medical staff (as is highlighted as a limitation of the study later on). 2. Within the Ethics and dissemination section, might it be that the authors want to go further than saying the findings will 'inform understandings of sustainable careers' but also point to some aspirations for demonstrable change as per the last objective of the study regarding making policy recommendations. Introduction 3. As per point 1 above, please include 'retention of doctors' as clarification in this section. 4. When giving details of the JLA PSP it would eb helpful to give some more introductory information about what JLA PSPs are and who they work with such that the reader can appreciate the importance of the outputs .i.e. Flag they include a range of stakeholders, produce a Top 10 etc. 5. Later in the methods section there is little description of how the team proposes to achieve objective 3. Further info within the methods section would be helpful. Methods 6. The study proposes to use participant observation as a method for data collection. Is the researcher (DD) part of the site clinical team who will be making judgements about patients care? If yes, this should be made explicit. If not, would this method then be considered non-participant observation? 7. The section on PPI and EM doctor input is important. It would be more helpful for the reader if the authors could include more information on whether and what changes to the study design these consultations produced. 8. How were the 2 sites for the observation work selected? It would eb helpful to provide some contextual information on the sites i.e. size of population they sere, type so EM admissions, size od department etc. Also, do the authors plan to explore what the retention rates for EM staff at these sites have been like over the past 5-10 years? This data would also be helpful int the analysis and to be able to consider how the findings form these sites do or do not transfer to others sites. A concern here is that if both sites are poor retainers, and that the study has set out to identify 'what drives retention of doctors in EM by focusing on those who stay in these careers', it may not be possible to achieve this as well as could be from other centres. 9. Would it be possible to select sites base don sites that retain well versus sites that do not? Or this sampling frame (a positive negative deviant approach) could be used to sample staff included in the interviews. 10. How have the areas of focus for the observation work (working environment, interaction between staff, and behaviours related to retention) been in informed? 11. How will the doctors who have left EM be identified and invited for interview? 12. Are there, or could there be, any plans to also include the GMC survey data s a method of triangulation across the data sets? Will the data from these surveys be used to inform the areas for investigation in interviews and observations? 13. Within the analysis section there isn't any detail on how the policy and literature documents will be analysed. For example will this be done concurrently with the observation/interview data, or sequentially and if so in what order? Also what method will be used, thematic analysis again or a directed content analysis? 14. Likewise there are no details on how objective 3 will be achieved
--	---

	with regard to method for development. Will all the data be brought together and summarised accordingly to target different audiences? Who will be involved in this process? How will the information be disseminated? Limitations 15. Include potential limitations of sites if relevant as per comments above.  • Authors may want to provide information on how this study has been funded.
--	---

REVIEWER	Deborah Swinglehurst QMUL UK
REVIEW RETURNED	28-Apr-2020

GENERAL COMMENTS	Thank you for asking me to review this protocol. I must begin with an apology as I have been unable to return this within the reviewer deadline in light of Covid-related disruption to my own work. I enjoyed reading the protocol and believe that an ethnographic study of the work of emergency doctors may certainly shed useful light on the lived experience of ED doctors. I have some areas of concern which I feel need to be addressed. I will refer to the relevant sections by line number. My key overall concern relates to the kind of data that is available and the issue of 'retention'. Whilst I appreciate that an ethnography will provide fascinating insight into the nature of ED work in its detail, the link between this and retention is not so clear. You don't 'observe' retention per se. This may seem like a statement of the obvious but it is not clear to me what constitutes e.g. 'retention-related behaviours' (L265) or how one 'keeps a focus on retention' (L197) - I think more thought needs to be given to what it is you CAN access / observe / analyse and how this might contribute to the debate and very real issue of retention. One can observe current work, working conditions, elucidate concerns etc, and these may very well be relevant to retention but it is not an observation/ethnography of 'retention'. There are no research questions in the protocol – it may be that clarifying the research question(s) in addition to the broad aim may help with this issue. In L 32 you refer to "those who stay" – but (related to above concern) we do not know if the people you observe are those that stay (longer term). Your ethnography can reveal the work of those who are currently working in the ED, and yes, you may get some insight into their satisfaction with their role, what they most value and their short term intentions, but it is a stretch to be confident that they are in fact 'those that stay'. However I absolutely agree that a study of the nature of the work can contribute something important over and above accounts of those who choose to leave the profession (though you do include some interviews with these people too, so be a bit more precise in your description of what you are doing). There is a discrepancy in your abstract between L 32 ("those who stay") and L 39 ("those who leave") which needs to be corrected. L 44/5 – suggest you give the date/number of your ethics / HRA approvals. L 123-4 You state that more experienced senior doctors deliver better care. Is this so? What is this assertion based on? L 135 I am not sure that the costs regarding replacement doctor (US) figure is relevant to a UK based study given the vast
---

	differences in the system/remuneration and would suggest you consider removing this. L143 – Formatting problem – the box is hiding some of the text I think? L150 – sentence needs recrafting Revisit Aims / Objectives (see comment at start of this review regarding what it is possible to find out). A relatively short period of observation will not allow you to assess sustainability, though of course may give some important insights. L 197 – I am not sure what ‘keeping a focus on retention’ would look like in practice. L198 – who are the ‘others’ who can inform the (missing) research question? This needs spelling out. The diagram on page 9 could be improved. The meaning of the arrows is unclear, and the linearity they suggest is possibly a bit misleading for work of this kind. L 259 – 267. You describe ‘half days’ but clearly the pattern of life in ED does not neatly fall into half days and it sounds like you plan to schedule observations at different times of day and night to accommodate different kinds of work experience. It would seem important to incorporate observations at all hours. A programme of observation of half day 4 days a week is very intense and I wonder if this is sustainable in such an information rich, fast moving environment. I speak from experience of conducting ethnographies in health settings and would struggle to sustain this intensity of observation. My experience of writing up field notes is that a 1:1 ratio of time is very optimistic. It may be a trade of between depth/richness/detail of observation and breadth. My personal preference is the former. In your shadowing it is not clear how you will deal with observation of patients (some of them acutely ill /very sick or distressed) with whom the doctors interact. In your shadowing how do you intend to manage this and how will you decide who and what is appropriate to observe (or not). L274 – You refer to DD’s existing knowledge. In the interests of reflexivity a statement of how DD comes to have ‘insider’ knowledge is needed, also some reflection on the relevance of this for access / analysis etc. L319 – Please include a statement about your sampling / sampling frame. Given your interest in retention I wondered why length of service as an ED doctor got no mention? At what point in a doctor’s career do you deem that they ‘have stayed’? L326-333 / 361-2 I presume you will have written consent at institutional level? Please make this clear. I was left a little unsure about what you meant by proportionate consent in this context? Do you mean processual consent or something different? You cannot observe or shadow without consent from the staff member; how this is documented is a separate issue – perhaps it is that latter that you refer to when you talk about ‘proportionate’? Is it a note in your fieldnotes for example? I would like to see much clearer discussion about the observation of patients and the nature of patient consent – it is not appropriate to say you “will not ignore them” (please delete this statement in favour of a much more considered statement of how you will attend to the very real ethical issue of observation of patients and how you intend to do this (or not) and what will be your approach to consent and to documenting ethnographic observations that include patients (and /or their records). I would also like to see whether/ how your observations will extend into non-clinical areas or ‘off-duty’ spaces e.g. staff rooms / coffee breaks etc as I imagine that it is in this context that you could encounter doctors’ reflections
--	--

	with their colleagues on their work experience. L382-385 Please include some consideration of whether/how/ to what extent you would seek to protect anonymisation of ‘elite’ interviewees and reflect on the consequences for your interviews if you are unable to offer anonymization. L 389 + Interviews / Policy documents are not mentioned at all in your analysis – how will you deal with this data and how does it relate to your observations? Please indicate interview type e.g. semi-structured / in-depth Appendix Interview Guide I have some minor suggestions on framing questions. Opening. What attracted you to work in emergency medicine? I was unsure what the question about “can you talk about a specific experience” related to? And the following Q regarding ‘what affected your decision’ was similarly unclear. Typo Ques – Cues Please explain what you mean by ‘using a social media post to generate discussion’? I wondered if calling it a Topic Guide as my sense is that you wish to adapt this iteratively. I hope these suggestions for improvement of your protocol are helpful and wish you well with your interesting study.
--	---

VERSION 1 – AUTHOR RESPONSE

Comment	Response
Reviewer 1	
In the third paragraph you talk about a broad issue in staffing, but when I read statistics about GPs, it seemed to muddy the waters for me. I think it would be more clear to stick to what’s happening in ED, unless you are making a case that the findings of this study may have implications for staff retention in General Practice.	This example was included to make the point that staffing is an issue across the health care economy. However, by removing the example this strengthens the focus and the brevity may even help reinforce the point
Line 125 should have a reference after “specialty”—although I see the following sentences are referenced, it would be helpful to also reference that broad claim.	You are quite right, the references in the following sentences support that statement. As such I have placed them after specialty as you suggest.
I am curious about your use of “participant” observation. This is indeed the traditional term but I think what you are describing is non-participant observation. I see that the researcher doing the data collection, the ethnographer, is also an ED doctor, unless they are also acting as a doctor in this setting during observation, I would have thought the term was non-participant observation.	The term I use to describe the practice of being in the space, observing, and recording is an interesting one, and one on which there are differing interpretations. On one hand I am not participating in my clinical role, however ‘non-participant’ suggests that I am passive in the space. I would contend that instead I am participating, but as a researcher, and by using this term I make it explicit that I expect that this presence in some way influences the observation. In this way I don’t see participant-observation as a dichotomy, or even a linear scale, but as a descriptive term. I am both observing the space, and participating in the space – but as a researcher. This is an interesting debate and one that I hope to engage with – however I don’t think

	the protocol has space for this, and I am therefore opting to retain the term as it acts as a useful shorthand. In the end, the actual description of what happens when I write the account of the research will describe what participant-observation ends up being, and how this compares with what I envision it to be is in itself interesting, but again probably not for the protocol. I think this reflects one of the core problems with ethnographic research, it is used by many different disciplines and each has developed its own overlapping lexicon. I don't think that the terminology that I have chosen is right, but I think it is the best fit for this study.
You note further down (lines 205-6) that an ethnographic approach is open-ended, but on line 197 you say that speaking to doctors in ED, the focus will be kept on retention. This struck me as not very ethnographic because at the heart of ethnography is the idea that you might go in to a setting/culture and find connections that you didn't expect, that might not jump out at you at first, so it's important to keep the inquiry broad, even with tight timelines. Perhaps it's just something to consider. I think it also becomes really clear that people often say one thing and do another, and this is found in ethnographies everywhere!	We acknowledge that going into a setting will lead to unexpected insights. However, this study is inspired by the 'problem oriented' ethnographic studies in healthcare. These are 'focused ethnographies', which are shorter and more intensive, and look at things like hip fracture anaesthesia and what makes a very safe maternity unit. While the approach is open ended and broad, focusing on the (still very broad concept of) retention as the 'problem' will allow for insights to be relevant for policy and practice rather than this being a general study of 'what it is like to work in the ED'. Ethnographic studies in EDs, or any location specific ethnography, can be the more traditional type looking at the environment in totality and trying to understand a something important about it. Or the more recent, more targeted ethnographies, with recent examples looking at CPR decision making (10.1016/j.socscimed.2016.03.022) or leadership (10.1136/medhum-2018-011517), still inductive in nature, but much more focused, what Pink describes as 'short-term ethnographies' (10.1002/symb.66). This is what I am trying to say as succinctly as possible. An ethnography could try and look on everything, but this study aims to look only at retention and I cannot state exactly what it will look at as that is not in keeping with the inductive nature of the study.
In line 240 you use the term "sense-checking" and I wondered if that is a particular method or just a phrase? If the former can you please reference and expand? Or perhaps reword for clarity?	Yes, that is a bit of jargon from the patient and public involvement world. I have changed it to 'reviewing'.
You mention on line 296 "pilot interviews" but I didn't see them mentioned above, did I miss it?	The pilot interviews will be with emergency physicians from outside the research sites. The

Will these pilots be an early subset of participants outlined above, or others?	data will not be used in the study except to hone the interview guide and iron out any practical problems. I don't think expanding on the detail of the pilot interviews is a good use of limited space in the manuscript so I have not done so here.
TYPO: line 314 "interviews candidates"- "s" at end of interview should be deleted	Thanks. Corrected.
I was interested that the inclusion criteria is "staff members for at least 4 weeks" given the introduction describes the benefits of experienced staff. I wonder why the criteria wasn't for staff who had been working for much longer so they could reflect on actual experiences, rather than expected futures?	It is certainly the case that more experienced staff would be included, such as consultants with many years of experience. However I thought it was also important to include rotating EM trainees who may have been an EP for several years but just rotated into the department as their relative newness means that they are likely to notice differences and distinctions about the department that more established members no longer notice. Also while less experienced doctors may not have been 'retained' for the same length of time, they are some of the people who the specialty need to retain.
Line 354, It's not that clear what is meant by this sentence, starting with "Capacity..." As in staff members might not have full faculties (as in be unconscious?)? Seems so unlikely it feels odd to mention.	A statement to this effect was required for HRA approval, but I agree that it is not required for the readers of the BMJOpen. As such I have deleted the paragraph.
TYPO? Line 355 "is" at the end of the line should read "in"	This paragraph has now been deleted.
Regarding consent, it does seem very unlikely but I wonder if there are plans for if a staff member is opposed to ever being observed and how that will impact general observation in that facility? This isn't an ethics application so it might not need to be included here, but it might be worth noting as it's high stakes for this study.	Yes this was explored in more detail in the ethics application, but as you say it is very unlikely. Essentially if it was a time limited objection – please don't observe at this time – the researcher would move to a different part of the department. If it wasn't time limited then the researcher would not attend when that person was on shift, the 24-hour nature of EM makes this feasible. This is summarised in the article: When introducing himself in the research setting DD will reiterate the voluntary nature of the study and that it is ok for them to ask for observation to stop, either temporarily or until further notice. Based on previous studies in the ED this is unlikely to occur frequently, and as such we intend, if this happens, to relocate to a different part of the department to continue observation. And we have added If the objection is not time limited then the researcher would not attend when that person was on shift, the 24-hour nature of EM makes this

	feasible.
Again, it's not an ethics application but in my experience it is not possible to allow people to withdrawal from the research all the way up to publication (line 384) since their data has already long before been collated and used to form the basis of analysis.	This is a long-standing recommendation from the ethics committee who approved the study. While these data will have been used to formulate broad categories of analysis, all effort will be made to withdraw direct quotations if requested by a participant.
Reviewer 2	
Abstract 1. Throughout the abstract it would help the reader if when mentioning retention that the term staff I also included. Also, here and throughout the protocol to would be helpful to highlight that the staff being referred to are doctors and not other medical staff (as is highlighted as a limitation of the study later on).	The first two paragraphs of the abstract already have 'doctors' after the term 'retention' so I have left these as is. The second paragraph in the introduction utilises the term 'medical staffing crisis' – quoting another paper (ref 4) so this cannot be changed. The first paragraph under the subheading 'from recruitment to retention' has the term 'registrar doctors' in the final sentence. The start of the 3rd paragraph refers to clinicians rather than doctors – this is deliberate – I think that while the evidence I draw on to make the argument that more experienced clinicians provide better care is mainly based on doctors, that it can be applied to other clinicians (advanced nurse practitioners for example) who do the work of seeing, assessing and managing patients. I think the rest of that paragraph accurately reflects the predominance of the literature in focusing on doctors in this regard, whilst trying to balance the reality that much emergency care is delivered by other professional groups of clinicians. The paragraph prior to 'research aim and objectives. Here I start without clearly defining the professional group in the first sentence, but do in the third. I have reorganised the paragraph so that the term 'doctors' is included at the outset. The opening of overview of the methods section contains the phrase 'EM doctors' which I think provides enough clarity for the remainder of that section. And, as you mention, the professional background of the interview candidates is again highlighted in the inclusion criteria.
2. Within the Ethics and dissemination section, might it be that the authors want to go further than saying the findings will 'inform understandings of	As specified in the abstract, the findings will have significant transferability as well as informing understandings.

sustainable careers' but also point to some aspirations for demonstrable change as per the last objective of the study regarding making policy recommendations.	I would argue that if the study aimed to facilitate demonstrable change then I need to measure that change. We don't have a good measure for retention rates and while I'm 100% on board that this is something that should happen, this isn't part of this study. While recommendations will be made and the success of the study is held accountable to funders and the public, I think what is written is a more achievable framing of the aim.
Introduction 3. As per point 1 above, please include 'retention of doctors' as clarification in this section.	See above.
4. When giving details of the JLA PSP it would be helpful to give some more introductory information about what JLA PSPs are and who they work with such that the reader can appreciate the importance of the outputs .i.e. Flag they include a range of stakeholders, produce a Top 10 etc.	Figure 1 summarises the JLA.
5. Later in the methods section there is little description of how the team proposes to achieve objective 3. Further info within the methods section would be helpful.	Within the ethics and dissemination section we have included the statement 'Policy briefs summarising the findings will be provided for key audiences such as NHS England, Health Education England, and the Royal College of Emergency Medicine.'
6. The study proposes to use participant observation as a method for data collection. Is the researcher (DD) part of the site clinical team who will be making judgements about patients care? If yes, this should be made explicit. If not, would this method then be considered non-participant observation?	No, DD is not a member of the clinical team making judgement about patient care. Reviewer 1 made a similar point. My response is copied below. The term I use to describe the practice of being in the space, observing, and recording is an interesting one, and one on which there are differing interpretations. On one hand I am not participating in my clinical role, however 'non-participant' suggests that I am passive in the space. I would contend that instead I am participating, but as a researcher, and by using this term I make it explicit that I expect that this presence in some way influences the observation. In this way I don't see participant-observation as a dichotomy, or even a linear scale, but as a descriptive term. I am both observing the space, and participating in the space – but as a researcher. This is an interesting debate and one that I hope to engage with – however I don't think the protocol has space for this, and I am therefore opting to retain the term as it acts as a useful shorthand. In the end I think the actual description of what happens when I write the account of the

	research will describe what participant-observation ends up being, and how this compares with what I envision it to be is in itself interesting, but again probably not for the protocol. I think this reflects one of the core problems with ethnographic research, it is used by many different disciplines and each has developed its own overlapping lexicon. I don't think that the terminology that I have chosen is right, but I think it is the best fit for this study.
7. The section on PPI and EM doctor input is important. It would be more helpful for the reader if the authors could include more information on whether and what changes to the study design these consultations produced.	None of the PPI led to huge changes in direction, it was more that it moulded a project over the 2 or so years it was in development. I think the one thing that I hadn't intended initially that is now part of the plan is the exit interview – this was initially suggested with professional PPI and further when the project received academic peer review. I have included the following to try and make this transparent. Including interviews with doctors who have left was suggested during this professional engagement, and again during peer review from the NIHR when funding was being obtained, and subsequently included in the research plan.
8. How were the 2 sites for the observation work selected? It would be helpful to provide some contextual information on the sites i.e. size of population they serve, type of EM admissions, size of department etc. Also, do the authors plan to explore what the retention rates for EM staff at these sites have been like over the past 5-10 years? This data would also be helpful in the analysis and to be able to consider how the findings from these sites do or do not transfer to other sites. A concern here is that if both sites are poor retainers, and that the study has set out to identify 'what drives retention of doctors in EM by focusing on those who stay in these careers', it may not be possible	This will be presented with the results for two reasons. First what the departments look like, who they see and how they work, has been in fairly constant change for the decade I have worked in the region – as such I think presenting what they are like at the time is more useful. Secondly, while I hope the sites don't change their mind it is possible – and has occurred before – and I think the protocol has flexibility as it stands to accommodate this (hopefully unlikely) possibility. With this in mind the inclusion criteria for the sites included in the protocol is deliberately open. Adding retention rates would be a good inclusion to the study, but these data are not available. Of note HEE have some case studies in the pipeline of places that are really good at retaining staff – this type of document may prove helpful to my analysis. With respect to your query about the individual characteristics (and possible failings) of the sites – this will be considered throughout the interviews and the analysis. It will be really informative to ask

to achieve this as well as could be from other centres.	someone who says 'we have a real retention problem here' – so what keeps you working here and in the profession?
9. Would it be possible to select sites based on sites that retain well versus sites that do not? Or this sampling frame (a positive negative deviant approach) could be used to sample staff included in the interviews.	See above.
10. How have the areas of focus for the observation work (working environment, interaction between staff, and behaviours related to retention) been informed?	Mainly from the preliminary literature review as part of the funding application, which is being updated by the scoping literature review which is currently near completion – the protocol for which is referenced in the paper (38).
11. How will the doctors who have left EM be identified and invited for interview?	'Potential participants for interviews will be identified during the periods of participant-observation and the literature review and supplemented by recommendations obtained from interviewees.' As a practicing clinician, well-integrated into EM trainee networks, I will be able to draw on current colleagues' recommendations of former colleagues. I could describe it as snowball sampling but then I would have to define the term and I hope that the simple description above is clear.
12. Are there, or could there be, any plans to also include the GMC survey data as a method of triangulation across the data sets? Will the data from these surveys be used to inform the areas for investigation in interviews and observations?	Not as triangulation, but as ethnographic data in of itself. The EMTA survey is something I also have access to and the two can be integrated into the analysis.
13. Within the analysis section there isn't any detail on how the policy and literature documents will be analysed. For example will this be done concurrently with the observation/interview data, or sequentially and if so in what order? Also what method will be used, thematic analysis again or a directed content analysis?	The data will be analysed together, policy, interviews, field notes, as one. The analysis section is not explicit in this point. I have tried to rectify this by inserting 'of fieldnotes, interview transcripts and literature and policy documents' into the second sentence of the analysis section which now reads: Once data collection is complete, a more thorough, systematic analysis of fieldnotes, interview transcripts and literature and policy documents will be completed utilising reflexive thematic analysis.
14. Likewise there are no details on how objective 3 will be achieved with regard to method for development. Will all the data be brought together and summarised accordingly to target different audiences? Who will be involved in this process? How will the information be disseminated?	The summary in the ethics and dissemination part of the protocol for this aspect of the study is very brief. I think the exact process will depend on the results, but would certainly be written specifically for different audiences. The PPI group will be key

	here both in making the message clear, and increasing the potential impact. The study is fortunately funded by three different organisations who can all help with this, and the host University has an established programme to help researchers gain traction both in policy circles and wider impact which I am actively engaged in. I totally agree this is really important to ensure that the sites and finders get value from the study, however I am not sure that the protocol necessarily benefits from expanding on this further.
Limitations 15. Include potential limitations of sites if relevant as per comments above.	I have added the following statement. The study is based at two sites. There is no reliable objective way of stating if the sites have high or low retention, relative to the national picture. If they are extreme outliers it may be difficult for other sites to interpret the findings.
 • Authors may want to provide information on how this study has been funded. 	The funding statement is included before the references.
Reviewer 3	
There are no research questions in the protocol – it may be that clarifying the research question(s) in addition to the broad aim may help with this issue.	The inclusion of a research question would indicate a more deductive approach to the problem, we have opted for a research aim and objective in keeping with the inductive nature of the study.
In L 32 you refer to “those who stay” – but (related to above concern) we do not know if the people you observe are those that stay (longer term). Your ethnography can reveal the work of those who are currently working in the ED, and yes, you may get some insight into their satisfaction with their role, what they most value and their short term intentions, but it is a stretch to be confident that they are in fact ‘those that stay’. However I absolutely agree that a study of the nature of the work can contribute something important over and above accounts of those who choose to leave the profession (though you do include some interviews with these people too, so be a bit more precise in your description of what you are doing). There is a discrepancy in your abstract between L 32 (“those who stay”) and L 39 (“those who leave”) which needs to be corrected.	Yes, brevity (for the abstract) has got in the way of clarity here and has been distilled from the longer, and hopefully more clear argument that much of the existing research in this area has focused on people who leave the profession (or other professions). I think extending the sentence to include ‘where previous research has targeted those who have left’ and changing the word ‘stay’ here to ‘remain’. I hope this subtle change reflects that I know we can’t predict if they will stay, just that they have thus far. This also relates to a point about interviewing people at different stages in their careers – a senior consultant has chosen to remain for much longer than trainee, but remain they both have.
L 44/5 – suggest you give the date/number of your ethics / HRA approvals.	I would appreciate a steer from the editor on this - I have added the numbers, but the date doesn’t seem to be the style for this journal. Please advise if these dates are required. For reference the dates are: FHMREC18058 – 15th April 2019 HRA – 11th November 2019

L 123-4 You state that more experienced senior doctors deliver better care. Is this so? What is this assertion based on?	The references in the following sentences in the paragraph support this statement, however you are right it would be much clearer if I included them in support of this statement as well, which I have done.
L 135 I am not sure that the costs regarding replacement doctor (US) figure is relevant to a UK based study given the vast differences in the system/remuneration and would suggest you consider removing this.	There is no reliable estimate for this in the UK, however I think even at a fraction it is still clearly a significant cost, and I think this is a relevant point to make. I think, on balance it adds something useful to the overall argument that retention is an important topic for research.
L143 – Formatting problem – the box is hiding some of the text I think?	Formatting issue will be resolved when figures are uploaded separately, as requested by the editor.
L150 – sentence needs recrafting	I have edited it to include a clarification that I am talking specifically about doctors in EM.
Revisit Aims / Objectives (see comment at start of this review regarding what it is possible to find out). A relatively short period of observation will not allow you to assess sustainability, though of course may give some important insights.	See comments above as well. I agree about the relatively short period of observation, however I think the aim and objectives focus on understanding. Insights could be substituted for this, and while in the rest of the protocol I talk about the context of sustainability, I don't make the claim that the study will assess sustainability, though I do hope it can inform understanding of it.
L 197 – I am not sure what 'keeping a focus on retention' would look like in practice.	Ethnographic studies in EDs, or any location specific ethnography, can be the more traditional type looking at the environment in totality and trying to understand a something important about it. Or the more recent, more targeted ethnographies, with recent examples looking at CPR decision making (10.1016/j.socscimed.2016.03.022) or leadership (10.1136/medhum-2018-011517), still inductive in nature, but much more focused, what Pink describes as 'short-term ethnographies' (10.1002/symb.66). This is what I am trying to say as succinctly as possible. An ethnography could try and look on everything, but this study aims to look only at retention and I cannot state exactly what it will look at as that is not in keeping with the inductive nature of the study.
L198 – who are the 'others' who can inform the (missing) research question? This needs spelling out. The diagram on page 9 could be improved. The meaning of the arrows is unclear, and the linearity they suggest is possibly a bit misleading for work of this kind.	This detail is given later in the section under the heading interviews; for brevity, it is not duplicated. It does suggest a linearity that does not exist. Rather than changing the diagram, which to do as you suggest I think is beyond my ability to distil the concept into an image, I have expanded on the text supporting the image. It now reads Figure 2. Summary of the ethnography. The

	arrows suggest a linearity to the study which is not reflected in the reality of ethnographic practice, instead the image is aiming to convey the different workstreams that come together in the analysis. The figure also fails to include the messy connections between the different parts of the study and the overlap between the intended outputs.
L 259 – 267. You describe ‘half days’ but clearly the pattern of life in ED does not neatly fall into half days and it sounds like you plan to schedule observations at different times of day and night to accommodate different kinds of work experience. It would seem important to incorporate observations at all hours. A programme of observation of half day 4 days a week is very intense and I wonder if this is sustainable in such an information rich, fast moving environment. I speak from experience of conducting ethnographies in health settings and would struggle to sustain this intensity of observation. My experience of writing up field notes is that a 1:1 ratio of time is very optimistic. It may be a trade of between depth/richness/detail of observation and breadth. My personal preference is the former. In your shadowing it is not clear how you will deal with observation of patients (some of them acutely ill /very sick or distressed) with whom the doctors interact. In your shadowing how do you intend to manage this and how will you decide who and what is appropriate to observe (or not).	Yes that is correct, the plan is to observe at all times. You may be right, I have had advice in both directions. The balance may evolve as the study progresses. I don’t anticipate that the patient encounter will be a significant part of my observations. These have a significant literature base already that I can draw from. The study does not have a shadowing element, I will not be following ‘key informants’ for allotted periods of time, rather I will be positioning myself in the department where I can observe the work outside of the cubicle.
L274 – You refer to DD’s existing knowledge. In the interests of reflexivity a statement of how DD comes to have ‘insider’ knowledge is needed, also some reflection on the relevance of this for access / analysis etc.	Totally agree, however I think this is best placed in the paper containing the results rather than in the protocol. This is stated on line 216 in terms of positionality. For the protocol I have added the brief clarification around the existing knowledge in parenthesis. ... study due to DD’s pre-existing knowledge of EM and EDs (as a senior trainee in EM).
L319 – Please include a statement about your sampling / sampling frame. Given your interest in	I do not have a pre-determined sampling frame, as the interviewees will be determined during the

retention I wondered why length of service as an ED doctor got no mention? At what point in a doctor's career do you deem that they 'have stayed'?	observations, specifying who to include in advance may be limiting. Instead I aim to justify who is interviewed in the results. A spread of career lengths in desirable and likely, I agree, and important. But I don't think specifying for example 2 trainees with < 3 years and 2 with over 2 years, 2 'new' consultant et cetera is advantageous or in keeping with the methodological basis for the study. 'Staying' in EM is understood as an on-going achievement, rather than something that has been achieved at a given point in time. My focus is on the micro-level activities and daily interactions that make staying possible. I agree that is vital to impose timeframes when trying to understand this from a human resource management perspective for example, where similarly definitions of things like turnover and retention rates would benefit from clarity.
L326-333 / 361-2 I presume you will have written consent at institutional level? Please make this clear. I was left a little unsure about what you meant by proportionate consent in this context? Do you mean processual consent or something different? You cannot observe or shadow without consent from the staff member; how this is documented is a separate issue – perhaps it is that latter that you refer to when you talk about 'proportionate'? Is it a note in your fieldnotes for example? I would like to see much clearer discussion about the observation of patients and the nature of patient consent – it is not appropriate to say you "will not ignore them" (please delete this statement in favour of a much more considered statement of how you will attend to the very real ethical issue of observation of patients and how	By proportionate I mean in the consent is measured against the risk of the study. I think it would be disproportionate to aim to obtain written consent from everyone in the department, for example. I am mostly familiar with processual consent from action-research. In this context I am not aiming to enact change in action, so while I agree that there are parallels that is not precisely what I mean. There is a distinction between what I understand 'processual consent' to be and the consent in this study. Again, I agree that I could not observe a staff member without consent, I think if I was doing shadowing then we would have leaned towards written consent, but I am observing the ED. The people are part of it. A note in the field notes is probably what will happen when these observations develop into informal, unrecorded, conversations as is typical of the ethnographic method. The consent in this instance will involve a reminder of who I am and why I am there, a reminder of the study, and an invitation to talk or not. I think the first sentence in this paragraph particularly 'no patient identifiable data will be recorded' is key. I have recrafted the sentence to not require this phrase, it now benefits from being more concise

you intend to do this (or not) and what will be your approach to consent and to documenting ethnographic observations that include patients (and /or their records). I would also like to see whether/ how your observations will extend into non-clinical areas or 'off-duty' spaces e.g. staff rooms / coffee breaks etc as I imagine that it is in this context that you could encounter doctors' reflections with their colleagues on their work experience.	as well. We believe it is unfeasible, overly burdensome, and unnecessary to obtain consent from patients transiting through the ED, a point endorsed by HRA ethical approval. The study could explore non-clinical spaces, such as the break room. These would be managed much the same as the rest of the study and if it became apparent that this area should be off limits then I would leave the note pad in my bag. However, as I am spending a reasonably prolonged period of time in the departments I would imagine using the spaces as intended. I think this is covered in the statement 'At each site they will perform observations in all parts of the department'. Like who will be interviewed (discussed below), I am reluctant to be too prescriptive as it may be limiting. However, I do intend to recount where and why locations were chosen for observations.
L382-385 Please include some consideration of whether/how/ to what extent you would seek to protect anonymisation of 'elite' interviewees and reflect on the consequences for your interviews if you are unable to offer anonymization.	If the interviewee feels that anonymisation is not possible then they can choose either to participate or not on this basis. This will be discussed in advance of the interview if it is anticipated that anonymisation is impossible, or at any point when it becomes apparent that we cannot maintain anonymisation. The statement about having the right to have their interview anonymised relates to those for whom it is possible, but they decide not desirable. Should they change their mind to believe that it is now desirable to be anonymised then I will do this. If this is not possible the interview will be withdrawn.
L 389 + Interviews / Policy documents are not mentioned at all in your analysis – how will you deal with this data and how does it relate to your observations? Please indicate interview type e.g. semi-structured / in-depth	The analysis section does not refer to the type of document being analysed (be that policy, interview transcripts or field notes). It is all data. The table, I hope, makes this clear with state 1 highlighting this. The interviews could be characterised as semi-structured, as the interview guide does impose some structure. I think they are more appropriately described as ethnographic interviews, given that the content will depend on the observations undertaken that have led me to select that individual to be invited for interview. The problem I have with calling them this in the protocol is that it does not draw a distinction between the talk that occurs during observation,

	which some authors call ethnographic interviewing, and the formal recorded interviews which require consent. With these two things in mind I felt that the clarification gained by calling the interviews 'semi-structured' or 'ethnographic' would be far outweighed by the potential confusion over the terminology. I think this reflects one of the core problems with ethnographic research, it is used by many different disciplines and each has developed its own overlapping lexicon. I don't think that the terminology that I have chosen is right, but I think it is the best fit for this study.
Appendix Interview Guide I have some minor suggestions on framing questions. Opening. What attracted you to work in emergency medicine? I was unsure what the question about "can you talk about a specific experience" related to? And the following Q regarding 'what affected your decision" was similarly unclear. Typo Ques – Cues Please explain what you mean by 'using a social media post to generate discussion'? I wondered if calling it a Topic Guide as my sense is that you wish to adapt this iteratively. I hope these suggestions for improvement of your	I really appreciate these suggestions. The ethical approval for the study was explicit in allowing for updates to the interview guide as the study progresses and I interpret that as including this peer review. Thanks - I think this is closer to natural speech. This is to try and draw together the interviews and the ethnography. If I can get the interviewees to talk about specific experiences, these may be observable. The second part, about what affected the decision, if to push the other way, to try and get the interviewees to talk about things that are not observable. Typo – corrected The working environment of the ED is a fairly recurrent topic for discussion on social media, particularly twitter. I keep track of relevant posts. Should the interview dry up I could access these, show them to the interviewee and ask for their thoughts. This would aim to generate discussion in the abstract, as the posts are either anonymous or about a department somewhere else. This abstract discussion could then lead to more concrete local examples. As the document is mainly for internal use, the titles main purpose is to define is as relating to the interviews apart from the participant-observation.

protocol are helpful and wish you well with your interesting study.	
---	--

VERSION 2 – REVIEW

REVIEWER	Dr Katie Gillies Health Services Research Unit, University of Aberdeen
REVIEW RETURNED	01-Jul-2020

GENERAL COMMENTS	Thank you to the team for providing responses to reviewer comments. I understand the argument given for the use of the term participant observation. However, given that two reviewers believe this term should be changed to non-participant observation (largely to enable the reader to grasp what is being done) I would suggest the team consider this change. The discussion about terminology and the overlapping meaning of these terms could be explored in a results/discussion paper but I think tis is important to change in the Protocol.
---

REVIEWER	Deborah Swinglehurst QMUL
REVIEW RETURNED	09-Jul-2020

GENERAL COMMENTS	The authors have made some modest changes to the manuscript but I do not feel that they have addressed all my comments adequately in the paper itself. I have not made any attempt to assess the extent to which I feel the other reviews' comments have been dealt with. I think this is potentially publishable but still does need some further refining. In my first review I set out one key overall concern as follows: "My key overall concern relates to the kind of data that is available and the issue of 'retention'. Whilst I appreciate that an ethnography will provide fascinating insight into the nature of ED work in its detail, the link between this and retention is not so clear. You don't 'observe' retention per se. This may seem like a statement of the obvious but it is not clear to me what constitutes e.g. 'retention-related behaviours' (L265) or how one 'keeps a focus on retention' (L197) - I think more thought needs to be given to what it is you CAN access / observe /analyse and how this might contribute to the debate and very real issue of retention. One can observe current work, working conditions, elucidate concerns etc, and these may very well be relevant to retention but it is not an observation/ethnography of 'retention'. There are no research questions in the protocol – it may be that clarifying the research question(s) in addition to the broad aim may help with this issue" I do not feel that the authors have engaged with this key concern in the revised paper and it seems important. I thought that a steer towards identifying research questions might help in this regard. If not I would still like to see some evidence that this concern is addressed. When I ask (L 197) what does 'keeping a focus on retention look like' in practice the response is a justification for the 'focus'. I appreciate that this is a focused ethnography and the
---

authors offer some suggestions of other focused ethnographies in making the point that a focused ethnography is what is planned. However whilst I can see how one might focus on CPR decision making (by observing and interviewing about CPR practices) and leadership (by observing occasions when leadership is happening) I remain confused about HOW one focuses an ethnography on retention by observing staff at work in an A+E department. I understand that the first author is familiar with the A+E environment (it is a long time since I was a clinician in A+E) and I think this is an issue of just reframing. As I say at the outset, I have no doubt that ethnographic approaches can offer important insights, but I would really like a clearer idea of what kinds of things the authors plan to observe in the pursuit of this focus on retention.

I pointed out what seems like a discrepancy in the abstract – in line 32 you say the study focuses on those who ‘stay’ in their careers. But in line 39 you say that you will interview doctors who leave. I realise these are not mutually exclusive but it remains a bit confusing to the reader.

The figure – with the explanatory footnote - is an improvement Regarding the period of ethnographic observation and my query about the intensity of the programme suggested, if there are plans to adapt the programme of observation iteratively (as is often the case in this kind of work) it would be sensible to include a sentence to this effect. A protocol sets out what you will do, and if some room is needed for iterative adaptation of some aspects of it then it may be sensible to incorporate something to this effect. This will appease people of a more quantitative leaning for whom the protocol is the script of what exactly you will do!

Observations. When I read the protocol first time round I had understood (wrongly perhaps!) that with a focus on doctors in the A+E department that observation in all parts of the A+E department would incorporate observing them doing their clinical work as this is what I assume they are doing most of the time – hence my question about how you will deal with the issue of patients who come into your field of observation. I assume many readers may make similar assumptions and it sounds like this is incorrect. In this case I think it would be VERY helpful to make it explicit that you do not intend to observe clinical work/interactions with patients and use this opportunity to explain what kinds of activity you DO plan to observe. This may also help with my overarching question of what a ‘retention related behaviour’ looks like and what you do intend to focus on. I realise that there are other kinds of activity in a department and perhaps I have been mistaken in my assumptions here. When you position yourself in the department what do you plan to observe? I know you cannot know what you can see before you go there but what kinds of interest do you have in observing? I think some greater clarity on this would be enormously helpful in this paper. You later state in your response that you will be focusing on ‘micro-level activities and daily interactions that make staying possible’. Are you more interested in inter-professional interactions than doctor-patient interactions? If so then it would be helpful to make this clear in the paper.

Regarding sampling. Whilst you may not be able to specify in advance precise numbers you could explain how you intend to go about your sampling (e.g. is this a purposive sample and if so with what interests in mind etc)

Thank you for clarifying your approach to consent and making some alterations to this section of your paper.

Interviews: Thank you for responding to my query about what kind of interviews you intend to conduct. From what you have written it

	sounds like you will be doing some formal audiorecorded interviews, which you describe as ethnographic interviews informed by a topic guide and adapted on the basis of the observations in each site (which is a perfectly adequate description and could be included in your protocol) alongside informal interviews (unrecorded, more spontaneous) I think it would be good to make this clear. It was not my intention that this should be a constraint, but the protocol is an opportunity to set out the different kinds of interview you plan to undertake and to be upfront / transparent about this. I think this is an important study, and it clearly already has the backing of NIHR. I do think the protocol could still benefit from some refining before publication.
--	--

VERSION 2 – AUTHOR RESPONSE

Comment	Response
Reviewer 2	
Thank you to the team for providing responses to reviewer comments. I understand the argument given for the use of the term participant observation. However, given that two reviewers believe this term should be changed to non-participant observation (largely to enable the reader to grasp what is being done) I would suggest the team consider this change. The discussion about terminology and the overlapping meaning of these terms could be explored in a results/discussion paper but I think tis is important to change in the Protocol.	We accept that the language is problematic but feel the term non-participant observation carries the same risks of misunderstanding as participant observation. Non-participant observation implies that the presence of the lead researcher in the environment will in no way influence it. Indeed, the choice of reflexive thematic analysis as an analytical process is deliberate in taking into account this inevitability. We, the research team, feel strongly that the term participant observation is the best fit to describe this element of the planned study. The term contains two descriptive terms of the activity of the researcher. At some points the researcher will be an active participant in the environment, but as a researcher not a clinician, speaking to people, at other he will be more focused on observation, on noticing. The glossary at the end of the paper contains a definition for participant-observation, which we have expanded to enable the reader to accurately understand the practice of ethnographic observation in this study.
Reviewer 3	
The authors have made some modest changes to the manuscript but I do not feel that they have addressed all my comments adequately in the paper itself. I have not made any attempt to assess the extent to which I feel the other reviews' comments have been dealt with. I think this is potentially publishable but still does need some further refining.	We thank the reviewer for pressing us on this point. It has helped us see where our thinking and rationale needed further explication. As the reviewer notes, retention isn't something you can see - only leaving is. On a day by day basis, people make a decision about whether to stay in a job or whether to leave it. But there isn't value in looking at the moment of leaving when trying to

In my first review I set out one key overall concern as follows:
“My key overall concern relates to the kind of data that is available and the issue of ‘retention’. Whilst I appreciate that an ethnography will provide fascinating insight into the nature of ED work in its detail, the link between this and retention is not so clear. You don’t ‘observe’ retention per se. This may seem like a statement of the obvious but it is not clear to me what constitutes e.g. ‘retention-related behaviours’ (L265) or how one ‘keeps a focus on retention’ (L197) - I think more thought needs to be given to what it is you CAN access / observe /analyse and how this might contribute to the debate and very real issue of retention. One can observe current work, working conditions, elucidate concerns etc, and these may very well be relevant to retention but it is not an observation/ethnography of ‘retention’. There are no research questions in the protocol – it may be that clarifying the research question(s) in addition to the broad aim may help with this issue”

I do not feel that the authors have engaged with this key concern in the revised paper and it seems important. I thought that a steer towards identifying research questions might help in this regard.

If not I would still like to see some evidence that this concern is addressed. When I ask (L 197) what does ‘keeping a focus on retention look like’ in practice the response is a justification for the ‘focus’. I appreciate that this is a focused ethnography and the authors offer some suggestions of other focused ethnographies in making the point that a focused ethnography is what is planned. However whilst I can see how one might focus on CPR decision making (by observing and interviewing about CPR practices) and leadership (by observing occasions when leadership is happening) I remain confused about HOW one focuses an ethnography on retention by observing staff at work in an A+E department. I understand that the first author is familiar with the A+E environment (it is a long time since I was a clinician in A+E) and I think this is an issue of just reframing. As I say at the outset, I have no doubt that ethnographic approaches can offer important insights, but I would really like a clearer idea of what kinds of things the authors plan to observe in

understand retention - that is the moment of understanding what pushed someone out of a job (or what pulled them to something else). Retention itself is a cumulative process, a lived experience that tests those push-pull factors in the actual context of work. While you can also argue that it relates to other factors (e.g. they might want to leave but can't afford to), understanding the everyday process of 'not leaving' is what is so interesting and what we seek to observe, because by looking at it, we can start to understand how we might shape the system to enhance the factors that make people not leave - aka have a long and sustained career in a difficult profession, which presents better value for money for medical training and provides better patient care - to facilitate greater retention. This point is further addressed below.

As mentioned in the previous response this study has a research aim and objectives, and not a closed research question.

This may very well mean observing, and noting, rather mundane things. Brief conversations between colleagues about work or non-work topics, gestures, kindness, humour. And things that make these things difficult, such as the physical space when it is ‘crowded’ or too hot or lacking in natural light. Or how people work around, or with, these barriers to continue with the ‘retention-related behaviours’. How they build a day-to-day process of not leaving in the context of the emergency department, what tactics they use to make work sustainable.

The challenge to your question is placing such a description in a protocol without pre-empting findings from this study and keeping such a description extremely succinct, as is required. We have, however, added a sentence in the middle of the paragraph you refer to, as below.

This ethnographic study will involve the lead researcher spending time in two different EDs,

the pursuit of this focus on retention.	trying to understand what allows doctors to have sustainable careers. This will involve observing the people, space, and happenings in the departments, and speaking to those in these spaces, keeping the focus on retention... Without aiming to predict retention-related behaviours, this may mean observing humour, conversations between members of staff, or how they work within the challenging working environments. ...This will be supported by interviews with doctors (conducted away from the department) and with others who can inform the research aim and objectives, and by critical review of policy and academic literature relating to retention. If the editors allow a slight increase to the word length, we would be happy to expand on this point.
I pointed out what seems like a discrepancy in the abstract – in line 32 you say the study focuses on those who 'stay' in their careers. But in line 39 you say that you will interview doctors who leave. I realise these are not mutually exclusive but it remains a bit confusing to the reader.	This statement is referring to the existing body of research which is primary focused on people leaving the profession. The choice to interview people who have left is to offer an alternative viewpoint, to challenge the ongoing analysis, in a not dissimilar way as choosing to interview policy makers. Unfortunately, there is insufficient room to expand on this in the abstract, and on reflection the rationale for including this group is not clear within the body text. We state, at the end of the section on patient and public involvement 'Including interviews with doctors who have left was suggested during this professional engagement, and again during peer review from the NIHR when funding was being obtained, and subsequently included in the research plan.' We have added, in the section on interview below this, a sentence to outline what I hope to gain from interviewing this group. ...), and ten with doctors who have left EM. The interviews with doctors who have left will still focus on retention and will provide opportunity to challenge the ongoing analysis. Potential participants for interviews will be identified...

The figure – with the explanatory footnote - is an improvement Regarding the period of ethnographic observation and my query about the intensity of the programme suggested, if there are plans to adapt the programme of observation iteratively (as if often the case in this kind of work) it would be sensible to include a sentence to this effect. A protocol sets out what you will do, and if some room is needed for iterative adaptation of some aspects of it then it may be sensible to incorporate something to this effect. This will appease people of a more quantitative leaning for whom the protocol is the script of what exactly you will do!	We have included a sentence to this effect, as you suggest at line 203. The study is inherently iterative, observations and interviews will inform subsequent observations and interviews, and the study plan may adapt to analysis, and opportunities and challenges in the field.
Observations. When I read the protocol first time round I had understood (wrongly perhaps!) that with a focus on doctors in the A+E department that observation in all parts of the A+E department would incorporate observing them doing their clinical work as this is what I assume they are doing most of the time – hence my question about how you will deal with the issue of patients who come into your field of observation. I assume many readers may make similar assumptions and it sounds like this is incorrect. In this case I think it would be VERY helpful to make it explicit that you do not intend to observe clinical work/interactions with patients and use this opportunity to explain what kinds of activity you DO plan to observe. This may also help with my overarching question of what a ‘retention related behaviour’ looks like and what you do intend to focus on. I realise that there are other kinds of activity in a department and perhaps I have been mistaken in my assumptions here. When you position yourself in the department what do you plan to observe? I know you cannot know what you can see before you go there but what kinds of interest do you have in observing? I think some greater clarity on this would be enormously helpful in this paper. You later state in your response that you will be focusing on ‘micro-level activities and daily interactions that make staying possible’. Are you more interested in inter-professional interactions than doctor-patient interactions? If so then it would be helpful to make this clear in the paper.	The sentence added above, in response to the required clarification on how I will observe things related to retention, starts this discussion, as you suggest. I do not plan on observing the clinical interactions directly, setting aside the additional burden of ethical approvals, we believe this aspect of medical work is documented and understood, by the anticipated audiences of this research, to a much greater degree than the continuing of the work outside of the direct clinical encounter. Most of the emergency physicians time is not spent in the face-to-face consultation, but in the work around these consultations, communicating with others in the department to ensure treatment or care is delivered, communicating with others outside of the department with actions such as referral and gaining specialist opinions, and indeed in the production of written communications in the form of clinical notes and discharge summaries. There is the additional work of getting background information through electronic health records and by calling care homes and general practitioners and family members. For the audiences of this research, we

	think the direct-clinical encounter is the given in the work of an emergency physician, some are easy, some are difficult, some go well, some badly, where think this study can add value, and understanding, is by understanding how is managed by the actors, the emergency physicians, outside the microcosm of the clinical encounter, through interactions with other members of the team. And how the work environment influences this. Again, the challenge here is producing a succinct summary of this, without, as you acknowledge, pre-empting the observation and analysis. Hopefully the specific example of 'humour' and highlighting the anticipated importance of 'conversations between members of staff', goes someway to helping the reader grasp what the lead researcher might be observing. Helped by the statement in the participant-observation section where we state 'We will investigate the working environment, interaction between staff, and behaviours related to retention' which should be clearer now that we have added a sentence clarifying what 'keeping focus on retention' means. We are, therefore, more interested in inter-professional interactions than the doctor-patient interaction. To try and communicate this succinctly, we have added the following to the paragraph below the sub-heading participant-observation. We will investigate the working environment, interaction between staff, and behaviours related to retention... Clinical encounters are not the focus of this study, they have been studied from multiple perspectives and can be considered a 'taken for granted' component of the study. ...The field notes will be handwritten The existing literature, limited though it is, does not point to clinical work in terms of understanding retention. It is more about rotas, shift patters, teamwork, being or feeling un/supported, all things that we expect to be able to observe outside of the cubicle.
Regarding sampling. Whilst you may not be able	As indicated above, the sampling approach will be

to specify in advance precise numbers you could explain how you intend to go about your sampling (e.g. is this a purposive sample and if so with what interests in mind etc)	guided by the field work, in, as I have added in response to your comment above, an iterative fashion. And that they will be identified 'during the periods of participant-observation and the literature review and supplemented by recommendations obtained from interviewees'.
Thank you for clarifying your approach to consent and making some alterations to this section of your paper.	
Interviews: Thanks you for responding to my query about what kind of interviews you intend to conduct. From what you have written it sounds like you will be doing some formal audiorecorded interviews, which you describe as ethnographic interviews informed by a topic guide and adapted on the basis of the observations in each site (which is a perfectly adequate description and could be included in your protocol) alongside informal interviews (unrecorded, more spontaneous) I think it would be good to make this clear. It was not my intention that this should be a constraint, but the protocol is an opportunity to set out the different kinds of interview you plan to undertake and to be upfront / transparent about this.	We state, 'In addition to the participant-observation conversations described above, we will conduct interviews, reserving the term for the physically separate, planned, audio-recorded and transcribed encounters.' At no point do we use the term ethnographic interview in the manuscript.
I think this is an important study, and it clearly already has the backing of NIHR. I do think the protocol could still benefit from some refining before publication.	Thank you. We hope the revisions provide the required level of clarification.